# The Biodistribution of Replication-Defective Simian Adenovirus 1 Vector in a Mouse Model

**DOI:** 10.3390/v16040550

**Published:** 2024-03-31

**Authors:** Juan Chen, Xiaojuan Guo, Xiaohui Zou, Min Wang, Chunlei Yang, Wenzhe Hou, Matvey V. Sprindzuk, Zhuozhuang Lu

**Affiliations:** 1NHC Key Laboratory of Medical Virology and Viral Diseases, National Institute for Viral Disease Control and Prevention, Chinese Center for Disease Control and Prevention, Beijing 100052, China; chenj_btmc@163.com (J.C.); guoxj@ivdc.chinacdc.cn (X.G.); zouxh@ivdc.chinacdc.cn (X.Z.); wangmin@ivdc.chinacdc.cn (M.W.); yangcl2210@163.com (C.Y.); houwz@ivdc.chinacdc.cn (W.H.); 2School of Public Health, Baotou Medical College, Inner Mongolia University of Science and Technology, Baotou 014040, China; 3Henan Chemical Technician College, Kaifeng 475008, China; 4United Institute of Informatics Problems, National Academy of Sciences of Belarus, 220012 Minsk, Belarus; computerscience_matvei@mail.ru

**Keywords:** adenovirus, simian adenovirus 1, gene transfer, administration route, biodistribution, bioluminescence imaging, real-time PCR, mouse model, vectored vaccine

## Abstract

The administration route affects the biodistribution of a gene transfer vector and the expression of a transgene. A simian adenovirus 1 vector carrying firefly luciferase and GFP reporter genes (SAdV1-GFluc) were constructed, and its biodistribution was investigated in a mouse model by bioluminescence imaging and virus DNA tracking with real-time PCR. Luciferase activity and virus DNA were mainly found in the liver and spleen after the intravenous administration of SAdV1-GFluc. The results of flow cytometry illustrated that macrophages in the liver and spleen as well as hepatocytes were the target cells. Repeated inoculation was noneffective because of the stimulated serum neutralizing antibodies (NAbs) against SAdV-1. A transient, local expression of low-level luciferase was detected after intragastric administration, and the administration could be repeated without compromising the expression of the reporter gene. Intranasal administration led to a moderate, constant expression of a transgene in the whole respiratory tract and could be repeated one more time without a significant increase in the NAb titer. An immunohistochemistry assay showed that respiratory epithelial cells and macrophages in the lungs were transduced. High luciferase activity was restricted at the injection site and sustained for a week after intramuscular administration. A compromised transgene expression was observed after a repeated injection. When these mice were intramuscularly injected for a third time with the human adenovirus 5 (HAdV-5) vector carrying a luciferase gene, the luciferase activity recovered and reached the initial level, suggesting that the sequential use of SAdV-1 and HAdV-5 vectors was practicable. In short, the intranasal inoculation or intramuscular injection may be the preferred administration routes for the novel SAdV-1 vector in vaccine development.

## 1. Introduction

Adenoviruses are non-enveloped viruses with the morphology of an icosahedron, and the virion packs a genome of linear double-stranded DNA of 26–48 kb in length. Adenoviridae consist of five genera, among which Mastadenovirus infects mammals [1]. Most of the knowledge of adenovirus originates from the study of human adenovirus C (HAdV-C2 or HAdV-C5). Some adenoviruses have been reconstructed as gene transfer vehicles by modifying the genome of the wild-type virus. Adenoviral vectors are widely applied in the basic research of biology, clinical tests of gene therapy, and development of vectored vaccines [2,3,4,5].

There are some successes of using adenoviruses as vaccines or vaccine vectors [4]. Wild-type HAdV-4 and HAdV-7 have been used as vaccines against acute recruit respiratory syndrome by oral administration since the 1970s [6]. Replication-competent HAdV-5 carrying the glycoprotein gene of the rabies virus in the E3 region was mixed in baits and could effectively immunize foxes and raccoons in the forests in North America [4,7]. Adenovirus-vectored vaccines against the Ebola virus or severe acute respiratory syndrome coronavirus 2 (SARS-CoV-2) have been also approved and used to inoculate susceptible populations recently [8,9,10,11,12]. Pre-existing immunity hampers the transduction efficiency of adenoviral vectors, decreases the expression of transgenes, and, finally, affects the immune response of vectored vaccines [13,14]. Therefore, interest has been attracted to develop adenoviruses from rare serotypes as gene transfer vectors to avoid the effect of pre-existing immunity against the commonly used HAdV-5 vector in humans [15,16,17,18]. On the other hand, different adenoviral types possess various characteristics in cellular tropism, which could play key roles in eliciting an immune response by influencing the expression and distribution of the target antigen [19].

The structure proteins on the virion determine viral tropism. The shell of an adenoviral virion is composed primarily of 240 capsomeres of hexon trimers (12 per triangular facet of the icosahedron) and 12 pentameric penton capsomeres at each vertex of the icosahedron [20]. For Mastadenovirus, 12 fibers protrude from the pentons, each a trimer of the fiber polypeptide. Fibers are the major ligand for the adenovirus to attach to and thus infect the host cell by recognizing and binding to the cellular receptors [21,22]. Different adenoviruses encode different fibers, and these fibers can bind to various cellular receptors, which provides a basic explanation on the molecular mechanism for diverse viral tropism. However, it was also found that a penton base, hexon, and even VI protein inside the virion played roles in virus infection [21,23,24]. In addition, administration routes can affect the infection outcome and virus clearance by influencing the spatial and temporal distribution of the virus.

Previously, we have constructed adenoviral vector systems based on simian adenovirus 1 (SAdV-1) [25]. Human beings have no pre-existing immunity against SAdV-1, suggesting the advantages of using SAdV-1 vector in human gene therapy or vaccine development. Moreover, SAdV-1 has two types of fibers on the virion, which is different from existing adenovirus vectors and might lead to changed cellular tropism in vivo [26]. Here, we constructed replication-defective SAdV-1 vectors carrying firefly luciferase and GFP reporter genes and attempted to track the distribution of the virus in a mouse model after intravenous, intranasal, intragastric, and intramuscular administration.

## 2. Materials and Methods

### 2.1. Cells, Plasmids, Viruses, Oligonucleotides, and Mice

HEp-2 (ATCC CCL-23), 293 (ATCC CRL-1573), and 293SE13 cells were cultured in Dulbecco’s Modified Eagle’s Medium (DMEM) plus 10% fetal bovine serum (FBS; HyClone, Logan, UT, USA) at 37 °C under a humidified atmosphere supplemented with 5% CO_2_, and regularly passaged every 3 or 4 days. 293SE13 is a 293 cell line expressing the E1B55K gene of SAdV-1, which is used to package SAdV-1 recombinant viruses [25].

pKSAV1-EG is an adenovirus plasmid containing the E1/E3-deleted SAdV-1 genome (NPRC2.3.308, https://www.nprc.org.cn/#/Adenovirus/HSAdVTwo, accessed on 30 March 2024) [25]. Plasmids of pAdEasy-1 and pShuttle-CMV constitute a vector system for constructing replication-defective HAdV-5 [27]. pGL4.17[luc2/Neo] was purchased from Promega (Madison, WI, USA). SAdV1-EG is an E1/E3-deleted SAdV-1 carrying the human EF1a promoter-controlled GFP gene, which was constructed and prepared previously in the laboratory [25]. The primers for Polymerase Chain Reaction (PCR) were chemically synthesized, and information related to these oligos is summarized in Table 1.

Six- to eight-week-old female BALB/c mice were purchased from Beijing Vital River Laboratory Animal Technology (Beijing, China and maintained under specific pathogen-free conditions. All experiments were carried out in strict compliance with the Guide for the Care and Use of Laboratory Animals of the People’s Republic of China and approved by the Committee on the Ethics of Animal Experiments of the Chinese Centre for Disease Control and Prevention.

### 2.2. Construction of Adenovirus Plasmids

The GFP coding sequence in the adenoviral plasmid of pKSAV1-EG was replaced with GFluc by restriction-assembly [25]. Briefly, fragments of firefly luciferase (Fluc) and GFP genes were separately amplified by PCR and fused to form a GFluc fragment by overlap extension PCR, in which the two reporter genes were linked with the Thosea asigna virus 2A peptide (T2A) coding sequence (Table 1). pKSAV1-EG was digested with the restriction enzyme SpeI, recovered, combined with the GFluc fragment, and subjected into a Gibson assembly reaction (NEBuilder HiFi DNA Assembly Master Mix, Cat. E2621, New England Biolabs, Ipswich, USA). The product was used to transform Escherichia coli TOP10 competent cells to generate the adenovirus plasmid pKSAV1-GFluc, which contained an E1/E3-deleted SAdV-1 genome and carried human EF1a promoter-controlled GFluc.

The HAdV-5 virus bearing GFluc was constructed by the traditional method of homologous recombination in bacteria [27]. The GFP and Fluc was amplified by PCR (Table 1), digested with KpnI/AatII and AatII/XhoI, recovered, and inserted into the KpnI/XhoI sites of pShuttle-CMV by restriction–ligation cloning of three fragments. The generated shuttle plasmid of pSh5-GFluc was linearized by PmeI, combined with the pAdEasy-1 backbone plasmid, and used to electroporate the Escherichia coli BJ5183 strain. The generated adenovirus pKAd5-GFluc was further amplified in and purified from Escherichia coli TOP10. pKAd5-GFluc carried CMV promoter-controlled GFluc.

### 2.3. Preparation of Recombinant Viruses

SwaI-linearized pKSAV1-GFluc was mixed with a jetPRIME reagent (Cat. 114-15, Polyplus-transfection, Illkirch, France) and used to transfect 293SE13 cells. Rescued SAdV1-GFluc viruses were further amplified in 293SE13 cells and purified with the CsCl ultracentrifugation method [28,29]. The purified virus was aliquoted and preserved at −80 °C in a buffer containing 10 mM Tris-Cl, 150 mM NaCl, 1 mM MgCl_2_, and 5% glycerol. Virus particle titer was determined by measuring the content of genomic DNA after proteinase K lysis (Qubit 1X dsDNA Assay Kit, Thermo Fisher Scientific, Waltham, MA, USA), where 100 ng of genomic DNA is equivalent to 2.98 × 10^9^ vp (viral particles) since a 32.8 kb viral genome has a molecular mass of 2.02 × 10^7^. An infectious titer was determined on 293 cells with the limiting dilution assay by counting GFP+ cells 2 days post infection [30]. HAdV5-GFluc viruses were similarly prepared, except that PacI was used for linearization and 293 cells were used for packaging.

### 2.4. Administration of Recombinant Viruses to Mice

Viruses were administrated to mice through intravenous, intranasal, intragastric, or intramuscular routes. Purified viruses were diluted in phosphate-buffered saline (PBS, Cat. 14190144, Gibco, Thermo Fisher Scientific) to desired concentrations just before inoculation. For intravenous administration, 100 μL viruses were injected to each mouse through one of the lateral caudal veins. For intramuscular administration, viruses were injected into the quadriceps femoris muscles of the hindlimbs with a volume of 50 μL for each hindlimb (100 μL in total). Mice were fasted for 6 h before intragastric and intranasal administration. For the intragastric route, viruses in 500 μL PBS were delivered to the stomach of each mouse via a gastric gavage needle. Before intranasal administration, mice were anesthetized by an intraperitoneal injection of tribromoethanol at a dose of 400 mg/kg based on body weight [31]; and then, 40 μL viruses were dropped slowly into the nostrils of each animal by using a pipette with a volume range of 10–100 μL.

### 2.5. In Vivo Bioluminescence Imaging and Mouse Tissue Collection

Before the imaging of mice, depilatory cream was applied to the chest, abdomen, and hind limbs by using cotton swabs; and five minutes later, the covering hair were removed by scrubbing the mice with cotton swabs dipped in water. D-Luciferin in PBS was intraperitoneally injected into mice at a dose of 150 mg/kg (Cat. 40902ES02, Yeasen Biotechnology, Shanghai, China). Thirteen minutes later, animals were anesthetized in an anesthesia machine containing 2% isoflurane and imaged in the IVIS Lumina Series III with the isoflurane anesthesia gas on (PerkinElmer, Hopkinton, MA, USA). Regions of interest (ROI) were created on the image, and total flux was measured.

Some mice were euthanized by cervical dislocation, and 13 types of organs or tissues were collected, washed once with cold PBS, weighed, fragmented, and frozen at −80 °C for luciferase activity assay and virus genome determination, including heart, liver, spleen, lung, kidney, brain, trachea, esophagus, stomach, small intestine, large intestine, Peyer’s patches, and quadriceps femoris muscles. Especially, the stomach, small intestine, and large intestine were cut open with ophthalmic scissors to remove the contents.

### 2.6. In Vitro Luciferase Assay

Fragmented tissues of 60–200 mg were transferred to 2 mL tubes, and two 4 mm and four 2 mm sterile stainless-steel grinding balls were added to each tube. The tubes were seated in a rack and frozen in liquid nitrogen for 1 min before being transferred to the chamber of a cryogenic grinder (JXFSTPRP-CLN, Jingxin, Shanghai, China), which was precooled to −18 °C. A processing program of 60 Hz-30 s/rest for 10 s/8 cycles was applied. The tubes were centrifugated at 10,000× *g* for 30 s, cell culture lysis reagent (CCLR) of 0.9 mL was added to each tube, and the homogenization was repeated at 4 °C by using the cryogenic grinder. After centrifugation at 10,000× *g* for 5 min (4 °C), the supernatant was transferred to a new tube and used for luciferase activity assay in a GloMax 96 microplate luminometer according to the manufacturer’s instructions (Luciferase Assay System, Cat. E1500, Promega). RLU represented relative luminescence units.

### 2.7. Quantification of Virus Genomic DNA by Real-Time PCR

Fragmented tissues of 30–100 mg were similarly homogenized by using the cryogenic grinder, except that a tissue lysis buffer for DNA extraction instead of CCLR was added for the second homogenization. A supernatant of 330 μL was used for tissue DNA extraction in an automated nucleic acid extraction system by using a genomic DNA extraction kit for animal tissues (Cat. ZTLKD and NP968-C, TIANLONG, Shaanxi, China). Extracted DNA of 2 μL was used as the template for TaqMan real-time PCR to absolutely quantify virus genomic copy numbers by detecting Hexon or GFP genes with primers and probes listed in Table 1.

### 2.8. Titration of Neutralizing Antibody against Adenovirus

Blood samples were retro-orbitally collected from mice, and sera were prepared for NAb titration. Recombinant SAdV-1 or HAdV-5 carrying the GFP reporter gene were mixed with serially-diluted sera and left to stand at room temperature for 1 h. HEp-2 cells were added, the expression of GFP was relatively quantified 40 h post infection by using a multimode plate reader (INFINITE M1000 PRO, Tecan Austria GmbH, Grödig, Austria). The highest dilution of human serum that could inhibit more than 50% activity of the virus was defined as the NAb titer. The details of this assay can be found elsewhere [25,32].

### 2.9. Immunohistochemistry

A method of immunohistochemistry was employed to identify the types of infected cells by detecting the expression of the reporter gene GFP after intranasal administration. A virus at a dose of 1 × 10^10^ vp in 40 μL PBS was used to infect mice intranasally as described previously. Mice were sacrificed 24 h post infection. Tracheas and lungs were collected and fixed in 10% neutral buffered formalin for 24 h. Tissues of about 3 mm thickness were dehydrated in a graded series of ethanol solutions, embedded in paraffin, and cut in a microtome to slices that were 4 μm thick. After deparaffinization and heat-mediated antigen retrieval, the slides were covered with 3% hydrogen peroxide for 30 min at room temperate to suppress endogenous peroxidase activity. After PBS washing and 3% BSA blocking, slides were incubated with anti-GFP rabbit polyclonal antibody overnight at 4 °C (1:300 diluted, Cat. BE2002, EasyBio, Beijing, China), washed three times with PBS, incubated with horse radish peroxidase (HRP)-conjugated secondary antibody for 30 min at 37 °C (Cat. ab6721, abcam, Shanghai, China), further washed three times with PBS, incubated in a DAB (3,3’-Diaminobenzidine) substrate for color development, rinsed with water for 5 min, counterstained with hematoxylin solution for 5 min, rinsed with water, decolorized in 1% acid alcohol for 1 s, blued with ammonia water for 10 s, rinsed with water, mounted with neutral balsam, and photographed under a light microscope.

### 2.10. Flow Cytometry to Detect the Infection of SAdV-1 to the Macrophages in a Mouse Liver and Spleen

Three mice were intravenously infected with SAdV1-EG at a dose of 1 × 10^10^ vp per mouse in 100 μL PBS via one of the lateral caudal veins. Another 3 mice were mock-infected with PBS. Twenty-four hours post infection, mice were anesthetized by an intraperitoneal injection of tribromoethanol at a dose of 500 mg/kg based on body weight [31]. Liver perfusion was carried out based on a two-step collagenase perfusion technique by using the inferior vena cava as the site for catheterization [33,34]. Mouse parenchymal hepatocytes were isolated by collecting and washing the liver cell pellet 3 times after centrifugation at 50× *g* for 3 min [33]. After the first centrifugation at 50× *g*, the cells in the supernatant were further pelleted by spinning at 650 g for 10 min; and the pelleted cells were resuspended in cold 17.6% OptiPrep (Cat. 1893, Serumwerk Bernburg AG, Bernburg, Germany), covered with a layer of 8.2% OptiPrep, and centrifugated at 1400× *g* for 30 min. Non-parenchymal cells (NPCs) at the interface between the two layers of the gradient medium were collected, washed, and used for flow cytometry assay [33]. Spleens were isolated, rinsed with PBS, and minced in culture media with plastic plungers from 5 mL syringes. Splenic leukocytes were collected after filtering the spleen suspension through a 70 µm cell strainer, lysing red blood cells, and washing.

NPCs or splenic leukocytes of 1 × 10^6^ were suspended in 100 μL PBS, mixed with 11 μL of diluted Zombie NIR marker (Zombie NIRTM Fixable Viability Kit, Cat. 423105, BioLegend, San Diego, CA, USA), stood for 20 min at room temperature in the dark, washed with PBS containing 1% FBS, resuspended, blocked with 2 μL of anti-Fc antibody (Purified Rat Anti-Mouse CD16/CD32, Cat. 553141, BD) for 10 min in ice, washed, resuspended, mixed, and incubated with 11 μL of APC-conjugated F4/80 monoclonal antibody (Cat. 17-4801-82, Thermo Fisher Scientific) for 30 min on ice in the dark, washed, resuspended in 0.5 mL PBS containing 1% FBS, and subjected to flow cytometry assay (BD FACSAria II, BD, San Jose, CA, USA). Mouse parenchymal hepatocytes were incubated with a Zombie NIR marker, washed, and similarly assayed with a flow cytometer [35]. A Zombie NIR marker is an amine-reactive fluorescent dye that is non-permeant to live cells, and therefore, it can help distinguish live and dead cells.

### 2.11. Statistical Analysis

The data were presented as mean ± s.d. (standard deviation) directly or after logarithmic transformation. One-way or two-way Analysis of Variance (ANOVA) followed by multiple comparison tests or Student’s *t*-tests were carried out to assess the statistical difference of the data. *p* < 0.05 was considered to be significant.

## 3. Results

### 3.1. Construction of SAdV-1 Vector Carrying Dual Reporter Genes of GFP and Firefly Luciferase

The firefly luciferase (Fluc) gene, coding sequence of the self-cleaving 2A peptide from the Thosea asigna virus (T2A), and the GFP gene were fused together by overlap extension PCR, and the adenovirus plasmid pKSAV1-GFluc was generated by using a method of restriction-assembly (Figure 1A) [25]. The recombinant virus of SAdV1-GFluc was rescued (Figure 1B), purified, and identified by restriction analysis of the virus genomic DNA (Figure 1C). The expression of GFP and Fluc in SAdV1-GFluc-infected HEp-2 cells were confirmed (Figure 1D–F). SAdV1-GFluc contained an E1/E3-deleted SAdV-1 genome with the human EF1a promoter-controlled exogenous gene of GFluc. HAdV5-GFluc served as a control here, and it was a replication-defective HAdV-5 carrying the CMV promoter-controlled GFP and Fluc genes linked with T2A.

### 3.2. Biodistribution of SAdV-1 Vector after Intravenous Inoculation

The strategies of intravenous administration were schematically shown in Figure 2A,B. After intravenous administration of SAdV1-GFluc through the tail vein at a dose of 5 × 10^8^ vp per mouse, bioluminescence imaging showed that the main luciferase activity was found at the upper part of the abdomen, where the liver and spleen were located (Figure 2C). As the time extended, the luciferase activity decreased rapidly. The total flux dropped about 30 times at the end of the first week and about 170 times at the end of the second. A second intravenous injection of SAdV1-GFluc with the same dose was given 3 weeks post the first; very weak luciferase activity was detected 1 day later, which was about 100 times lower than that detected at day 1 post the first injection. Moreover, it dropped to the background at day 2 post the second injection (Figure 2D). The serum NAbs were detected in all virus-infected animals 3 weeks post the first infection, and the titer kept going up after the second injection (Figure 2E).

The SAdV-1 target organs in a mouse were identified by in vitro luciferase assay. Thirteen organs or tissues were isolated from infected animals, and considerable activity was found in the liver, spleen, and lungs as early as 6 h post infection. The highest activity per unit weight was seen in the spleen at an early period (Figure 2F). Luciferase activity decreased dramatically in both the liver and spleen, which was in line with the results from live imaging. In contrast, the luciferase activity in the lungs went down relatively slow. However, the total activity in the lungs was still approximately 7.5 times lower than that in the liver 2 days post infection. The change trend of the copy number of virus genomic DNA was consistent with that of luciferase activity, except that the virus DNA kept accumulating in the liver from 6 h to 2 days post infection (Figure 2H,I). Notably, the luciferase activity was relatively stable in the lungs, although the copy numbers of the virus genome decreased significantly. It seemed that low-level expression of the exogenous gene saved some infected lung cells from clearance by the immune system, while a portion of viruses temporarily attached in the lungs without effective infection, which was finally degraded or transferred to another place by blood flow. Trace luciferase activity or the virus genome were also detected in the kidneys of some virus-infected animals.

### 3.3. Biodistribution of SAdV-1 Vector after Intranasal Inoculation

The inoculation schemes for intranasal administration of SAdV1-GFluc are shown in Figure 3A,B. The results of bioluminescence imaging are shown in Figure 3C,D. It can be seen that the cells in both upper and lower respiratory tracts were infected. The major luciferase activity was found in nasal cavities and was relatively stable in 2 weeks, while the signal in the lungs gradually became weak and faded at day 15 post infection (Figure 3C). The inoculation dose of 5 × 10^8^ vp per mouse was relatively low. Therefore, the majority of the viruses were absorbed in the upper respiratory tract, including the nasal cavity, and few viruses had the chance to enter the lungs. After the second administration, the luciferase activity came back again and reached a level as high as that detected after the first inoculation. However, it went down much faster, as it decreased 4.6 times in 5 days (Figure 3C,D). The titers of NAbs against SAdV-1 were minimal but detectable, and it did not go up significantly after the second administration (Figure 3E).

Among the 13 isolated organs or tissues, the highest luciferase activity was detected in the lungs (Figure 3F,G and so was the copy number of the virus genome (Figure 3H,I). Virus DNA was also detected in the trachea (Figure 3H). Since the mass of tracheas was small, tracheas were included only in virus DNA detection and excluded for luciferase assay. The tissues from the upper respiratory tract were not included in these assays due to the difficulties in isolation. Notably, minimal luciferase and virus DNA were detected in the brains of some virus-infected mice.

### 3.4. Biodistribution of SAdV-1 Vector after Intragastric Inoculation

The schemes for intragastric administration are shown in Figure 4A,B. The results of bioluminescence imaging showed that luciferase activity was detected in mouse abdomens at day 1 post infection (Figure 4C,D). The activity decreased very quickly and could not be detected 7 days post infection. Moreover, the highest activity detected at day 1 was much lower than that detected after intravenous or intranasal administration of the same dose of virus. A total of 2.5 times more virus was given 21 days post the first infection. The luciferase activity was detected one day later, and it was 3.4 times higher than that detected at day 1 post the first inoculation. Similarly, the activity decreased abruptly in the following days (Figure 4C,D). No NAbs against SAdV-1 was detected 3 weeks post the first or the second administration.

Trace amounts of luciferase or virus DNA were detected in the stomach, small intestine, large intestine, and Peyer’s patches of mouse digestive systems 6 h post administration (Figure 4E–H). They decreased very quickly and were lower than the limit of detection 2 days post infection, which might be a result from the rapid turnover of epithelial cells on the gastrointestinal tract.

### 3.5. Biodistribution of SAdV-1 and HAdV-5 Vectors after Intramuscular Inoculation

The scheme for intramuscular repeated administrations is shown in Figure 5A. Recombinant SAdV1-GFluc and HAdV5-GFluc were injected. Mice were divided into three groups, and each group was inoculated with recombinant viruses at three sequential time points. For example, SSH-L represented the group of a SAdV1/SAdV1/HAdV5-Low dose (Figure 5A). The results of bioluminescence imaging for the SSH-L group are representatively shown in Figure 5B. The bioluminescence signal was only seen at the injection sites. The data of total flux are shown in Figure 5C,D. After the first injection, the highest expression of Fluc was seen at day 1 post infection. The luciferase activity remained relatively stable for the low-dose groups of SSH-L and HHS-L within 7 days, decreased dramatically at day 15, and almost reached the background at day 21 for all groups. Notably, when SAdV1-GFluc was given at a dose of five times higher, the total flux was approximately five times stronger in the SSH-H group than that in the SSH-L group at day 1 or day 2 post infection. However, the total flux reached the same level at day 7 (Figure 5D). At day 63 post the first injection, the same types of viruses at the same dose were repeatedly given. The total fluxes for all groups were significantly lower at day 1 post the second injection than those at day 1 post the first. Furthermore, they decreased much faster and reached the background at day 8 post the second injection (Figure 5C,D). Unexpectedly, a higher dose of SAdV1-GFluc (SSH-H group) did not lead to a stronger expression of luciferase at days post the second injection (Figure 5D). At day 85 post the first inoculation, heterotypic viruses were intramuscularly injected into the three groups of mice (Figure 5A). Interestingly, the expression of Fluc at day 1 post the third injection reached the level of that at day 1 post the first injection of the same type of virus. For example, in the SSH-H group, HAdV5-GFluc was given at a dose of 1 × 10^9^ vp per mouse for the third injection; the total flux at day 1 post the third injection was at the same level of that measured at day 1 post the first injection in the HHS-L group. However, the luciferase activity decreased much faster for the third time (Figure 5C,D).

Serum NAbs against SAdV-1 or HAdV-5 were titrated after the first, second, and third infections (Figure 5E–G). It could be seen that the NAbs against the homotypic virus were elicited to a low level after the first infection, and they increased abruptly to a very high level after a boost infection of the homotypic virus. If a larger dose of SAdV1-GFluc was given (SSH-H group), the titers of generated NAbs against that virus could be higher (Figure 5E,G). The titers of NAbs against the heterotypic virus remained at the background level after the first and second infections. A heterotypic virus was used for the third infection. Although it was the first time for the mice to be infected with the heterotypic virus, NAbs against the corresponding heterotypic virus accumulated to a relatively higher titer in a short period (16 days post the third infection). These data suggested that NAbs against SAdV-1 had little cross-reaction with that against HAdV-5. However, advanced infection of one type of the viruses had some positive effect on the generation of NAbs against the other. Anti-GFP antibodies in these collected sera were also semi-quantified by immunofluorescence assay (Appendix A).

The scheme for studying adenovirus biodistribution by tissue isolation is shown in Figure 6A. Mice were divided into three groups and were intramuscularly injected with a low dose of SAdV1-GFluc (SAdV1-L), low dose of HAdV5-GFluc (HAdV5-L), and high dose of SAdV1-GFluc (SAdV1-H), respectively. Among the 13 isolated organs or tissues, the luciferase activity was mainly found in muscles isolated from the injection sites, and it decreased more than 50 times in 2 weeks (Figure 6B). Trace luciferase activity was also found in the livers and spleens 6 h or 1 day post infection (Figure 6C,D). The virus DNA was also mainly found in muscles isolated from injection sites. The copy numbers of virus DNA in these muscles decreased relatively more slowly when compared to the luciferase activity, and they were still detectable 28 days post infection (Figure 6E). A trace amount of virus DNA was detected in the livers, spleens, or lungs isolated from some of the infected mice 6 h or 1 day post infection (Figure 6F).

### 3.6. Identification of the Types of Infected Cells after Intranasal Administration

Tracheas and lungs were isolated and immunohistochemistry (IHC) staining was performed to detect the expression of GFP 1 day post intranasal administration of SAdV1-GFluc at a dose of 1 × 10^10^ vp per mouse. HAdV5-GFluc served as a control (Figure 7). For either HAdV5-GFluc- or SAdV1-GFluc-infected mice, GFP expression was found in cells that constituted the epithelial lining of the trachea and the bronchial tree, including ciliated cells and the Clara cells. The epithelia of the alveoli were almost free from DAB staining, especially for HAdV5-GFluc-infected mice. Interestingly, it could be seen that some positively-stained cells were distributed in a dotted pattern in alveolus areas in the lungs of SAdV1-GFluc-infected mice (Figure 7). These cells might be macrophages, considering the distribution pattern and the function of macrophages. However, this kind of DAB staining could hardly be found in the lungs of HAdV5-GFluc-infected mice.

### 3.7. Confirmation of the Infection to Macrophages in the Liver and Spleen after Intravenous Administration of Recombinant SAdV-1

Liver non-parenchymal cells, parenchymal cells (hepatocytes), and splenic leukocytes were isolated and assayed by flow cytometry one day post intravenous administration of the SAdV1-EG virus to mice (Figure 8A–C). SAdV1-EG carried the GFP reporter gene, and an allophycocyanin (APC)-conjugated antibody against the F4/80 molecule was used to label macrophages. Among liver non-parenchymal cells, 9.79% of the F4/80+ cells expressed GFP, while only 0.46% of the F4/80− cells were GFP-positive. Similarly, the proportions of GFP+ cells were 5.13% and 0.19% for F4/80+ and F4/80− cells in splenic leukocytes, respectively (Figure 8D). The proportion of GFP+ cells in hepatocytes was 0.62%. If we combined data here with those from the biodistribution experiments (Figure 2), it could be concluded that macrophages in the mouse liver and spleen played a pivotal role in purging SAdV-1 viruses from blood. Since the absolute number of hepatocytes was greater than that of liver non-parenchymal cells [33], it could be inferred that considerable hepatocytes were transduced by SAdV1-GFluc if a dose as high as 1 × 10^10^ vp per mouse was administrated intravenously (Figure 8B).

## 4. Discussion

By analyzing the sequences of the SAdV-1 genome and its fiber proteins, it is possible to infer its primary cellular receptors. SAdV-1 was isolated from the cultured kidney cells of a healthy cynomolgus monkey (Macaca fascicularis) for the first time in 1955 [26,36], and it was sporadically detected in monkey feces. SAdV-1 can be non-pathogenic or cause mild infectious diarrhea in monkeys [37,38,39]. The genome of SAdV-1 has the high homology to that of HAdV-52 (identity 95%), both of which belong to HAdV-G in classification. HAdV-G viruses possess two types of fibers on the virion. It was found recently that HAdV-52 virions contained equal amounts of short and long fibers: the terminal knob domain of the short fiber (fiber1) binds to sialylated glycoproteins, while the long fiber (fiber2) binds to the coxsackievirus and adenovirus receptor (CAR) protein [40,41]. CAR homologs have been found to be highly conserved in evolution [42], and the HAdV-5 vector can transduce mouse cells through mouse CAR (mCAR) [43,44,45]. The lengths of fiber1 or fiber2 proteins are the same for both SadV-1 and HadV-52 (363 or 560 amino acids, respectively). The similarity of fiber1 proteins is 90% (327/363), while the identity of fiber2 proteins is 99% (552/560) for these two adenoviruses, suggesting that the two types of SAdV-1 fibers might have identical cellular receptors to that of HAdV-52 fibers, respectively.

Here, we comprehensively investigated the biodistribution of a replication-defective SAdV-1 vector in an infected mouse model. Luciferase activity and viral DNA in organs or tissues were detected in this study. Of note, these two indicators are not necessarily consistent with each other. Here are some reasons. The same promoter can have variable activity in different cell types. Different viruses might have distinct fates after entering the same type of cells. A degraded virus in a lysosome can no longer express the transgene, but the viral DNA is still detectable before complete degradation.

After intravenous administration, most SAdV-1 viruses were absorbed by macrophages in the liver and the spleen, while a considerable amount of liver parenchymal cells (hepatocytes) were also transduced. However, the duration of the expression of the exogenous gene in the liver and spleen was relatively short, and the infection elicited a strong immune response against the vector, which might cause side effects or hamper the repeated usage of the vector.

Intragastric administration of the SAdV-1 vector could cause the digestive tract to be transduced locally. However, the expression of the transgene was very weak and lasted a very short time (1–3 days). No Nabs against the vector was detected, implying that the administration could possibly be repeated many times. In other studies, mice were given HadV-41, carrying the spike gene of the Middle East respiratory syndrome coronavirus (MERS-CoV) through the intragastric route more than 6 times [46], and serum NAbs against MERS-CoV could be successfully elicited (unpublished data), for which a similar explanation was provided here. It is inconvenient and costly to inoculate a vaccine so many times. Further investigation is deserved to determine if an enteric-coated or nanoparticle-packed formulation could improve the effectiveness of oral SAdV-1 vectored vaccines [6].

A transgene could be expressed moderately in the respiratory tract, and the expression was sustained for more than 2 weeks without a notable decrease if the SAdV-1 vector was intranasally administrated. The serum NAbs against the vector were low in titer, and it did not compromise the expression of the transgene in the beginning days after a second administration was carried out (Figure 3D). IHC results showed that both SAdV-1 and HAdV-5 transduced the cells lining the lumen of the respiratory tract, while SAdV-1 might infect more macrophages in the lungs (Figure 7). HAdV-5 has one type of fiber, which interacts with CAR molecules on the epithelial cells. Higher amounts of CAR-binding fibers on the virion might help retain HAdV-5 in the respiratory tract. In contrast, half the number of fibers (fiber2) on the SAdV-1 virion bind to CAR, and a lower affinity to the respiratory tract would cause more virions to be released into the alveoli, where SAdV-1 vectors had more of an opportunity to transduce macrophages. Interestingly, it was reported recently that the deepened and widened biodistribution in the lungs would result in a much-improved vaccine-mediated immunogenicity and protection against the target pathogen [47,48]. Further investigation is deserved to determine if SAdV-1 has advantages over other adenoviral vectors when intranasal administration is employed. Notably, a trace amount of SAdV-1 was found in the brain. Similarly, intranasal delivery of replication-defective HAdV-5 also led to a moderate level of gene transfer in the olfactory bulb. However, intranasal inoculation led to little or no viral dissemination of the viruses to the major region of the CNS, the brain [49].

Considering that the intramuscular route is the most commonly used strategy for vaccine inoculation, single, repeated, and sequential administrations of SAdV-1 or HAdV-5 were investigated in detail. The performance of SAdV-1 was close to that of HAdV-5 when single or repeated administrations were applied. The local expression of the transgene sustained nearly one week and then decreased dramatically in the following weeks. The elicited immunity against the homologous vectors hampered the repeated administration. The inhibition effect was even more significant in the high-dose group of SAdV-1. Sequential administrations of SAdV-1/HAdV-5 or HAdV-5/SAdV-1 still led to a high expression of the transgene at day 1 post the last inoculation. However, the expression of the transgene decreased very rapidly in the following days, suggesting that cross-reactive cellular immunity instead of cross-reactive serum NAbs existed between these two types of adenoviruses. It seemed that sequential administration of SAdV-1/HAdV-5 had a slight advantage over the sequential administration of HAdV-5/SAdV-1, considering higher luciferase activity in the SAdV-1/HadV-5 group. A trace or no expression of the transgene was detected outside of the injection sites. The data indicated that SAdV-1 could be an excellent vaccine vector as HAdV-5 was, and sequential inoculation could be an option for the two vectors.

The cellular receptor of HAdV-5 was found to be CAR more than three decades ago. However, the expression pattern of CAR cannot properly explain the distribution of HAdV-5 in a mouse model. The structure of virion proteins as well as many host factors contribute to the outcome of HAdV-5 infection [19,50,51]. For example, HAdV-5 could directly or indirectly bind to the following host molecules: several blood coagulation factors (FVII, FIX and FX), serum natural IgM, macrophage scavenger receptor-A (SR-A), SREC-1, the heparan sulfate proteoglycans (HSPGs), LDL receptor-related protein, and cytosolic protein tripartite motif containing-21 (TRIM21) [51]. On the other hand, besides fiber, virus structural proteins of hexon, penton base, and VI participate in host–virus interactions. SAdV-1 and HAdV-5 are classified into distinct adenovirus species. Protein structures or sequences on the virion surface are different between SAdV-1 and HAdV-5, which might lead to diverse virus–host interactions. While efforts have been made to modify the HAdV-5 vectors to meet the requirement of gene therapy or vaccine development [52], SAdV-1 might have the potential to offer new features.

In summary, we systematically investigated the expression of a transgene after intravenous, intragastric, intranasal, or intramuscular administrations of the SAdV-1 vector in a mouse model, and the efficient expression was detected for the intranasal and intramuscular routes. Furthermore, SAdV-1 is different from the conventional HAdV-5 vector in virus structural proteins. Human beings possess negligible pre-existing immunity against SAdV-1, which provides a good reason to use SAdV-1 vectors in situations where HAdV-5 is unqualified or sequential inoculation will lead to a stronger immune response.

## Figures and Tables

**Figure 1 viruses-16-00550-f001:**
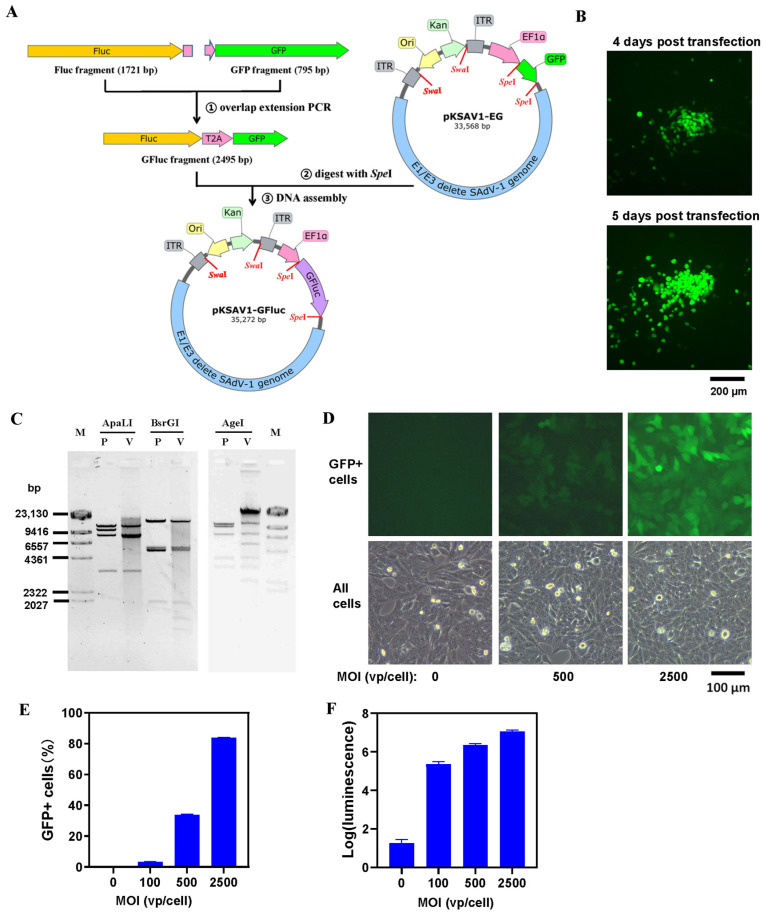
Construction and identification of a simian adenovirus 1 (SAdV-1) vector carrying the firefly luciferase and GFP genes. (**A**) Schematic diagram of constructing the adenoviral plasmid pKSAV1-GFluc. EF1a: human EF1a promoter, Fluc: firefly luciferase, ITR: inverted terminal repeat, T2A: self-cleaving 2A peptide from the Thosea asigna virus. (**B**) Rescue of the recombinant SAdV1-GFluc virus. Adenovirus plasmid pKSAV1-GFluc was linearized by SwaI digestion and used to transfect 293SE13 packaging cells. Foci formed by GFP-positive cells were found under a fluorescence microscope 4 days post transfection, indicating a successful virus rescue. (**C**) Identification of the SAdV1-GFluc genome by restriction analysis. Virus genomic DNA was digested with restriction enzymes and resolved on 0.7% agarose gel by electrophoresis. Plasmid pKSAV1-GFluc DNA was similarly processed and served as a control. The predicted molecular weight (bp) of the digested fragments of viral genomic DNA (V) was 3275, 8180, 8553, 12,771 for ApaLI; 1286, 1520,1923, 5313, 5645, 17,092 for BsrGI; 1440, 2161, 2871, 3500, 5021, 6937, 10,849 for AgeI. The predicted molecular weight (bp) of the digested fragments of viral plasmid DNA (P) was 3275, 8584, 10,642, 12,771 for ApaLI; 1923, 5299, 5313, 5645, 17,092 for BsrGI; 1440, 2871, 3500, 6937, 9675, 10,849 for AgeI. Human HEp-2 cells were infected with SAdV1-GFluc at MOIs of 0, 100, 500, and 2500 vp/cell for 6 h. At 48 h post infection, the expression of GFP was observed under a fluorescence microscope (**D**) or assayed by flow cytometry (**E**); and the activity of firefly luciferase was determined in a GloMax 96 microplate luminescence detector after mixing the cell lysate with the substrate (**F**).

**Figure 2 viruses-16-00550-f002:**
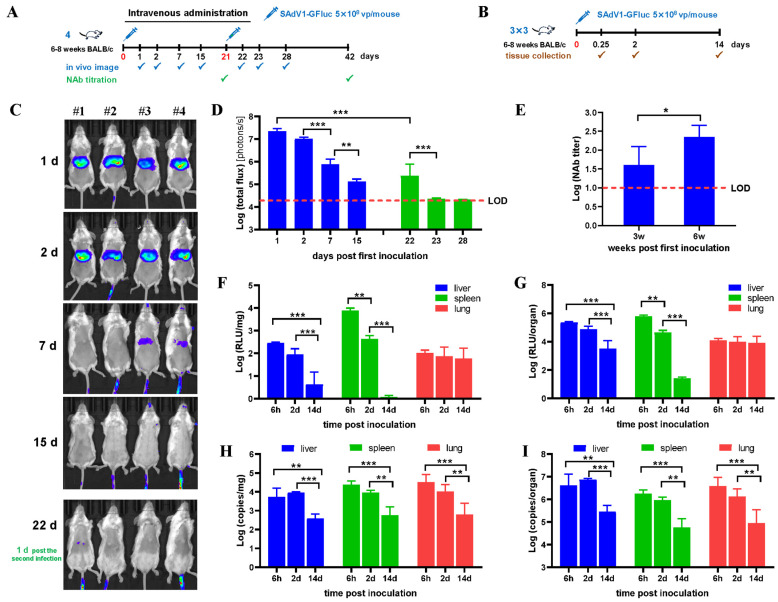
Biodistribution of the SAdV-1 vector in mice after intravenous administration. (**A**) Inoculation and detection scheme for live imaging and neutralizing antibody (NAb) titration. (**B**) Inoculation and detection scheme for in vitro detection of fire luciferase activity and virus genomic DNA. (**C**) Bioluminescence imaging to show the activity of firefly luciferase in mice on sequential days post the first virus infection. (**D**) Total photon flux measured after drawing the regions of interest (ROIs) on living images. (**E**) Blood samples were retro-orbitally collected 3 weeks post the first or the second virus infections, and sera were prepared and used to determine the titers of neutralizing antibodies (NAbs) against SAdV-1. Mice were sacrificed at indicated time points post the virus infection. Thirteen types of organs and tissues were collected and weighed. Total proteins were extracted and subjected into an in vitro luciferase assay (**F**,**G**) or total DNA were isolated and used as the template to determine the copy numbers of the SAdV1-GFluc genome by TaqMan PCR (**H**,**I**). Positive results were only detected in the organs of the liver, spleen, and lung, which are shown as per unit weight (**F**,**H**) or per organ (**G**,**I**), respectively. LOD: limit of detection. * *p* < 0.05, ** *p* < 0.01, *** *p* < 0.001.

**Figure 3 viruses-16-00550-f003:**
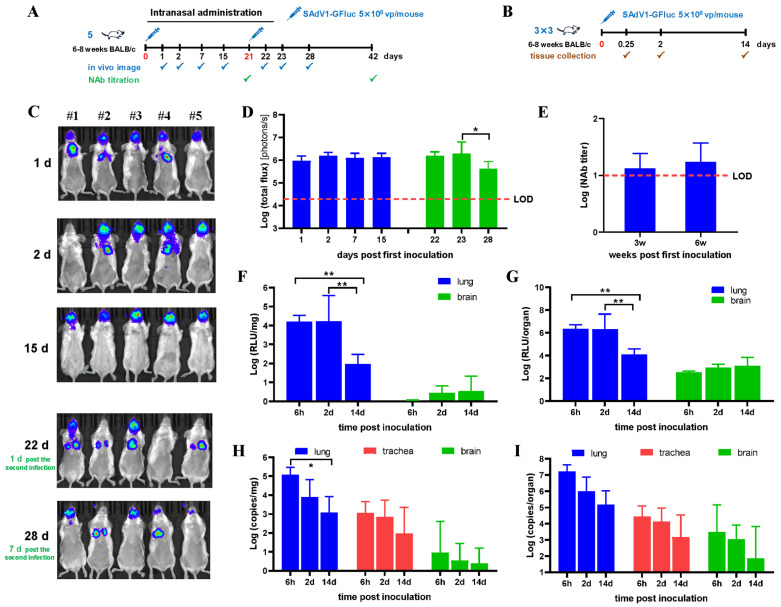
Biodistribution of the SAdV-1 vector in mice after intranasal administration. (**A**) Inoculation and detection scheme for live imaging and NAb titration. (**B**) Inoculation and detection scheme for in vitro detection of fire luciferase activity and virus genomic DNA. (**C**) Bioluminescence imaging to show the activity of firefly luciferase in mice on sequential days post the first virus infection. (**D**) Total photon flux measured after drawing the regions of interest (ROIs) on living images. (**E**) Blood samples were retro-orbitally collected 3 weeks post the first or the second virus infections, and sera were prepared and used to determine the titers of neutralizing antibodies (NAbs) against SAdV-1. Mice were sacrificed at indicated time points post the virus injection. Thirteen types of organs and tissues were collected and weighed. Total proteins were extracted and subjected into in vitro luciferase assay (**F**,**G**), or total DNA were isolated and used as the template to determine the copy numbers of the SAdV1-GFluc genome by TaqMan PCR (**H**,**I**). Positive results were only detected in the organs of the lungs, trachea, and brain, which are shown as per unit weight (**F**,**H**) or per organ (**G**,**I**), respectively. The mass of tracheas was very little and could hardly meet the need for luciferase assay. LOD: limit of detection. * *p* < 0.05, ** *p* < 0.01.

**Figure 4 viruses-16-00550-f004:**
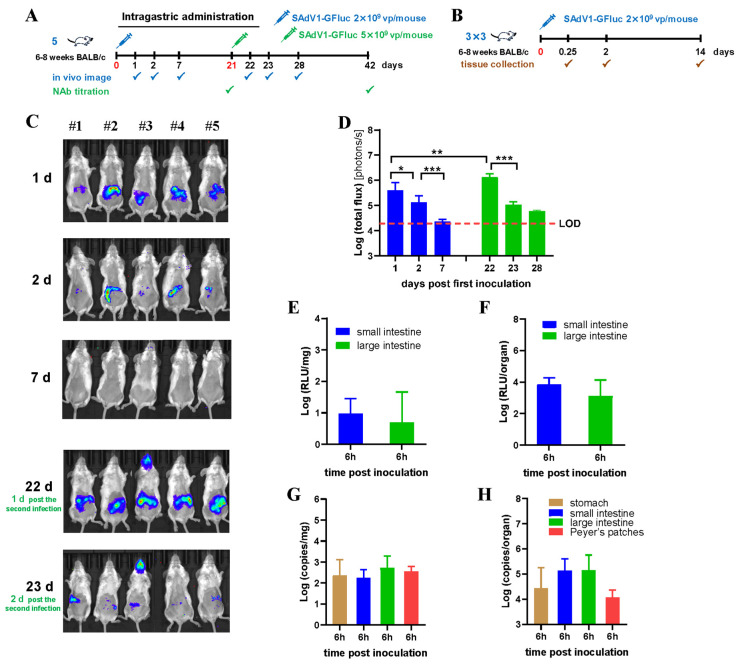
Biodistribution of the SAdV-1 vector in mice after intragastric administration. (**A**) Inoculation and detection scheme for live imaging and NAb titration. (**B**) Inoculation and detection scheme for in vitro detection of fire luciferase activity and virus genomic DNA. (**C**) Bioluminescence imaging to show the activity of firefly luciferase in mice on sequential days post the first virus infection. (**D**) Total photon flux measured after drawing the regions of interest (ROIs) on living images. Serum NAbs against SAdV-1 were detected as negative. Mice were sacrificed at indicated time points post the virus injection. Thirteen types of organs and tissues were collected and weighed. Total proteins were extracted and subjected into in vitro luciferase assay (**E**,**F**), or total DNA were isolated and used as the template to determine the copy numbers of the SAdV1-GFluc genome by TaqMan PCR (**G**,**H**). Positive results were only detected in the organs of the stomach, small and large intestines, and Peyer’s patches at 6 h post infection, which are shown as per unit weight (**E**,**G**) or per organ (**F**,**H**), respectively. The mass of the stomach and Peyer’s patches was very little and could hardly meet the need for luciferase assay. LOD: limit of detection. * *p* < 0.05, ** *p* < 0.01, *** *p* < 0.001.

**Figure 5 viruses-16-00550-f005:**
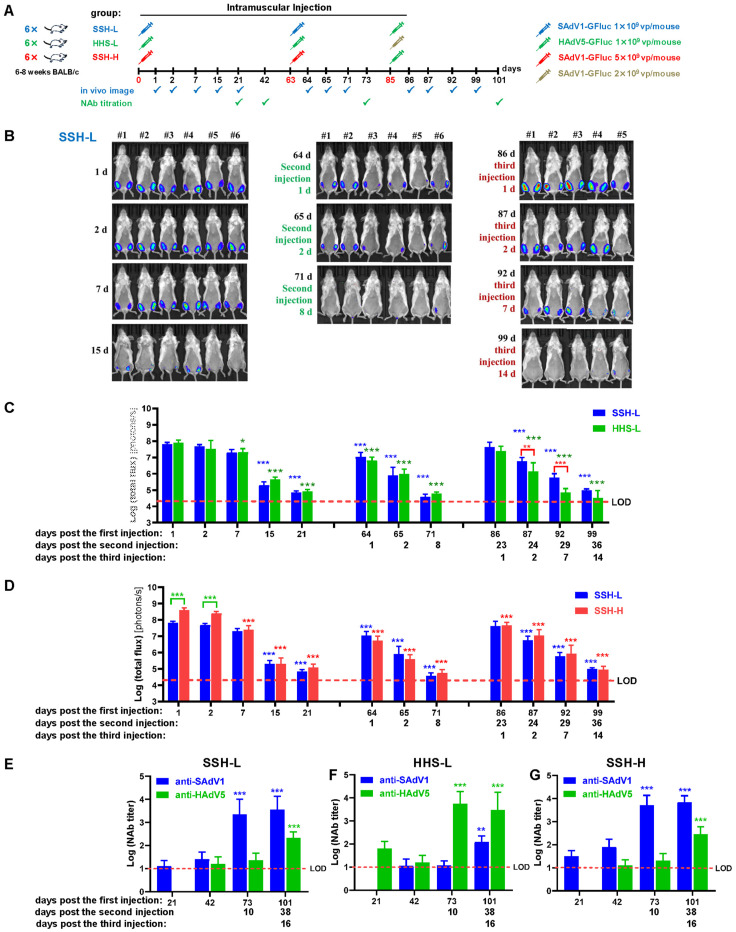
Biodistribution of SAdV-1 or HAdV-5 vectors in mice traced by bioluminescence imaging after intramuscular administration. (**A**) Inoculation and detection scheme for live imaging and NAb titration. (**B**) Bioluminescence imaging to show the activity of firefly luciferase in mice on sequential days post virus infection. Shown are the images from the SSH-L group. Total photon fluxes were measured after drawing the regions of interest (ROIs) on living images. The data from the SSH-L/HHS-L groups (**C**) and those from the SSH-L/SSH-H groups (**D**) were separately shown to facilitate a comparison. Serum NAbs against SAdV-1 or HAdV-5 were determined (**E**–**G**). LOD: limit of detection. SSH-L: SAdV1/SAdV1/HAdV5-Low dose group; SSH-H: SAdV1/SAdV1/HAdV5-High dose group; HHS-L: HAdV5/HAdV5/SAdV1-Low dose group. The data collected in the following days were compared to those collected the first, and statistical differences (*) were labeled on the top without connecting lines. * *p* < 0.05, ** *p* < 0.01, *** *p* < 0.001.

**Figure 6 viruses-16-00550-f006:**
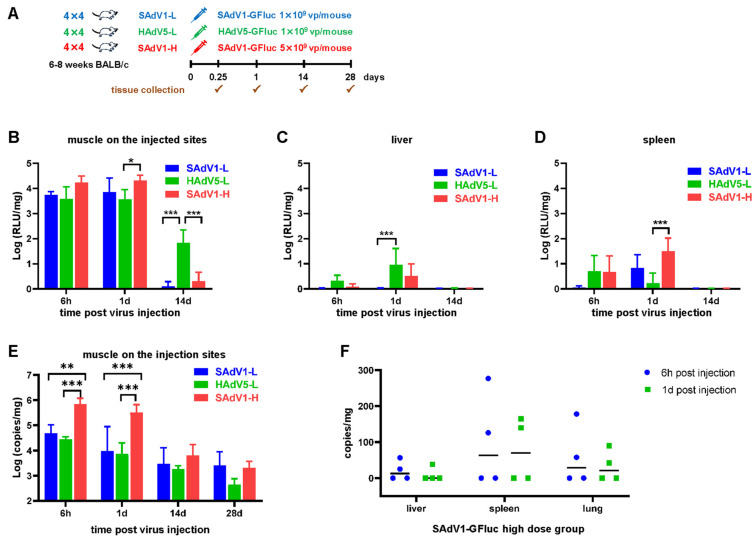
Biodistribution of SAdV-1 vector in mice by in vitro luciferase assay and virus DNA detection after intramuscular administration. (**A**) Inoculation and detection scheme. (**B**) Mice were sacrificed at indicated time points post the virus injection. Thirteen types of organs and tissues were collected and weighed. Total proteins were extracted and subjected into an in vitro luciferase assay, and positive results were only detected in the muscle (**B**), liver (**C**), and spleen (**D**). Simultaneously, total DNA were isolated and used as the template to determine the copy numbers of the SAdV1-GFluc genome by TaqMan PCR. Strong positive results were only detected in the muscle (**E**), while low copies of virus DNA were detected in the liver, spleen, or lungs of part of the infected animals in the SAdV1-GFluc high-dose group at 6 h or 1 day post injection (**F**). * *p* < 0.05, ** *p* < 0.01, *** *p* < 0.001.

**Figure 7 viruses-16-00550-f007:**
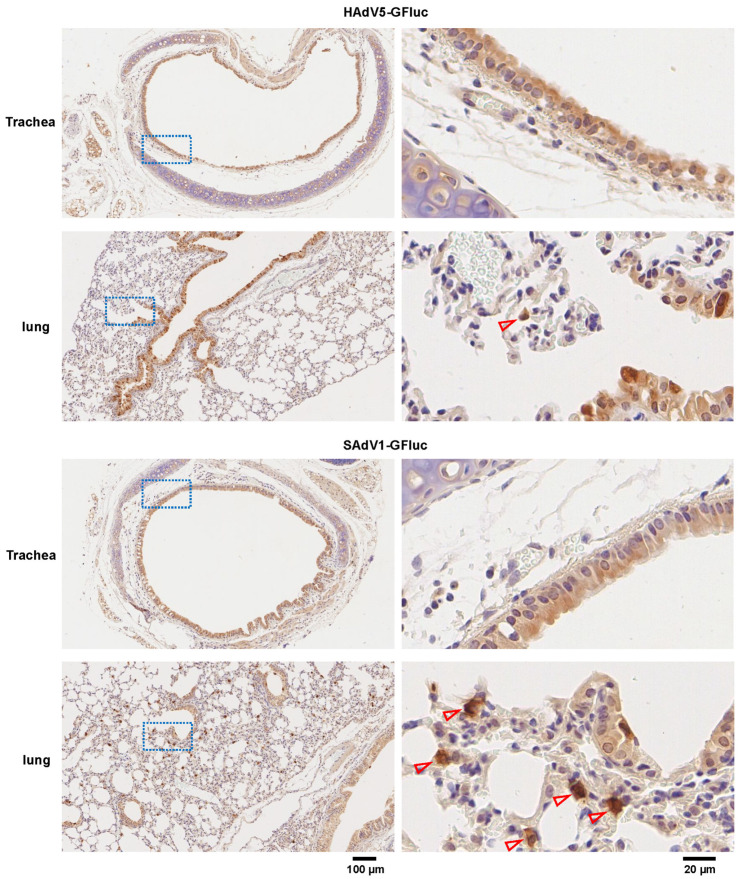
Detection of the expression of the GFP gene in the trachea and lungs by the immunohistochemistry (IHC) method 24 h post intranasal administration of HAdV5-GFluc or SAdV1-GFluc at a dose of 1 × 10^10^ vp per mouse. The brown signal represented the positive expression of GFP because DAB (3,3’-Diaminobenzidine) was used as the staining substrate. The details within the rectangular dashed area on the left were magnified and are shown on the right. Hollow arrowheads show positively stained cells in alveoli.

**Figure 8 viruses-16-00550-f008:**
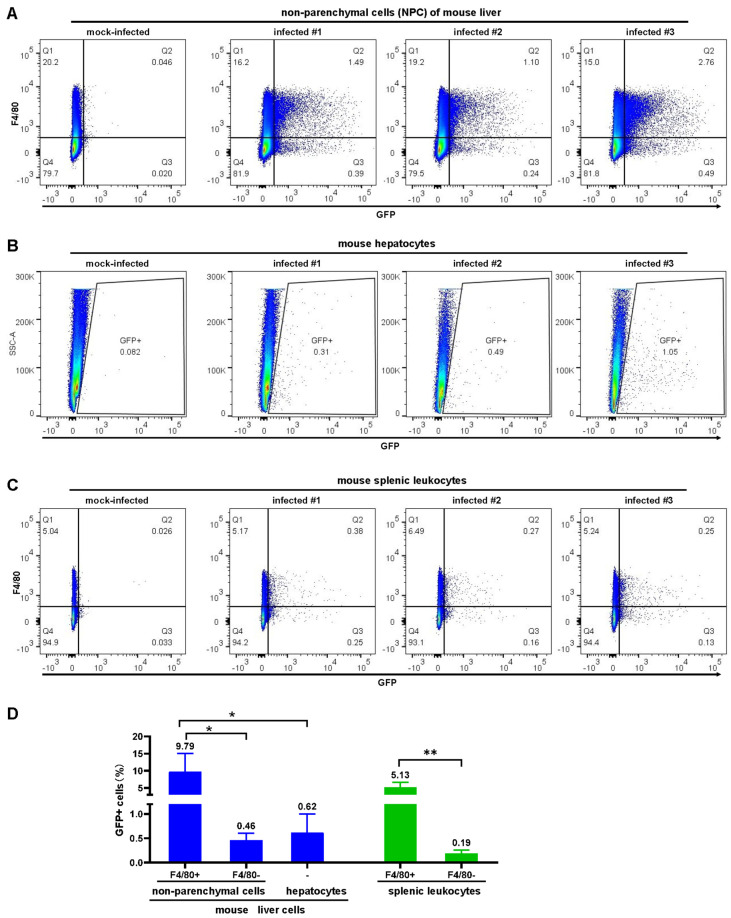
Infection of the SAdV-1 vector to macrophages in the mouse liver and spleen. Mice were intravenously inoculated with SAdV1-EG at a dose of 1 × 10^10^ vp per mouse. Twenty-four hours post infection, liver nonparenchymal cells (**A**), parenchymal cells (**B**), and splenic leukocytes (**C**) were isolated, stained with a Zombie NIR marker and APC-conjugated anti-F4/80 antibody, and assayed by flow cytometry. The proportions of GFP+ cells in the subgroups of F4/80+ or F4/80− leukocytes were calculated and subjected to statistical analysis (**D**). The F4/80 molecule is a typical marker of macrophages, * *p* < 0.05, ** *p* < 0.01.

**Table 1 viruses-16-00550-t001:** Summary information of oligonucleotides.

Fragment	Oligo Name	Sequence	Template	Product Length (bp)	Restriction Enzyme	Aim
GFluc-S	2103SAV1F1	ccaagctgtg accggcgcct acactagtgc caccatggaa gatgccaaaa acattaag	pGL4.17	1721	SpeI	construct SAdV1-GFluc
	2103SAV1R1	cgacgtcacc gcatgttaga agacttcctc tgccctccac ggcgatcttg ccgcccttc				
	2103SAV1F2	ttctaacatg cggtgacgtc gaggagaatc ccggccctat ggtgagcaag ggcgaggag	pKSAV1-EG	757		
	2003SAV2GFPr	ggtcaaggaa ggcacggggg agactagttt agagtccgga cttgtacagc tc		2495	SpeI	
GFluc-H	1403GFPT2A-F	ggccggtacc atggtgagca agggcgag	pLEGFP-C1	766	KpnI	construct HAdV5-GFluc
	1403GFPT2A-R	gccgacgtca ccgcatgtta gaagacttcc tctgccctcc ttgtacagct cgtccatgc			AatII	
	1403T2ALuc-F	ggccgacgtc gaggagaatc ccggccctat ggaagatgcc aaaaacatta ag	pGL4.17	1691	AatII	
	1403T2ALuc-R	ggccctcgag ttacacggcg atcttgccgc			XhoI	
Hexon-qPCR	2112SAdV1-Hexf1	gggctggttg acacctacgt	tissue DNA	118		detect SAdV-1 genome
	2112SAdV1-Hexr1	ccaggagcat ggaacggtag				
	2112SAdV1-Hexp	5′ FAM-caccaccgca acgccggact c-BHQ1 3′				
GFP-qPCR	2008GFPf	gacaaccact acctgagcac cc	tissue DNA	126		detect GFP DNA
	2008GFPr	cttgtacagc tcgtccatgc c				
	2008GFPprobe	5′ HEX-tccgccctga gcaaagaccc caac-BHQ1 3′				

## Data Availability

Data are contained within the article and Appendix A.

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
