# Peer review of "The Biodistribution of Replication-Defective Simian Adenovirus 1 Vector in a Mouse Model"

_viruses, 2024, doi:10.3390/v16040550_

Round 1
Reviewer 1 Report
Comments and Suggestions for Authors
In their manuscript titled “The Biodistribution of Replication-Defective Simian Adenovirus 1 Vector in Mouse Model”, Chen et al. provide data for the distribution of the virus after intravenous, intranasal, intragastric, and intramuscular administration. They provide an extensive dataset, detecting the expression of a transgene, quantifying viral genomes, and in the case of intranasal expression, attempting to identify infected cells by immunohistochemistry. The authors also experiment with schemes of repeated administration and measure neutralizing antibodies.
A major flaw of the manuscript that while the authors extol the vector’s applicability to vaccination, they only base this on transgene expression. To make this claim, they need to show the immune response to the transgene, and that repeated administration of the vector or alternating administration of SAdV-1 vector with HAdV-C5 vector is beneficial to vaccination with HAdV-C5 vectored vaccine only.
Specific criticism
1. In the figures showing in vivo luciferase expression and virus genome content, it is confusing to have the “per mg tissue” and “per organ” graphs. The “per mg tissue” graph is sufficient. Further, the statement that “the highest total activity (per organ) changed to liver at day 2” (lines 313-314) is irrelevant and needs to be removed.
2. In Fig. 2F and H, the graphs depicting luciferase expression and genome copy numbers are contradictory; luciferase expression is stable until Day 14 while the genome copy number declines. A similar discrepancy is shown in Fig. 6B and E at Day 14. The authors suggest that this is through “accumulation of luciferase protein and removal of viral DNA simultaneously” occurring (lines 322-323). As the half-life of the luciferase protein is measured in minutes, this is unlikely; another explanation is necessary.
3. Fig. 3 shows strong luciferase expression in the nose. This should be quantified, shown, and discussed.
4. The indication of 10-base logarithm is log, not lg; this needs to be changed in the graphs.
Comments on the Quality of English Language
Minor editing for syntax/clarity would be beneficial.
Reviewer 2 Report
Comments and Suggestions for Authors
In this manuscript, the authors generated recombinant simian adenovirus vector 1 harboring GFP and firefly luciferase (SAdV1-GFluc) and extensively examined biodistribution and transgene expression from the virus after intravenous, intranasal, intragastric and intramuscular administrations in a mouse model. They found that the virus efficiently expressed the transgene equivalent to frequently used human adenovirus vector HAdV5-GFluc in intramuscular inoculation. In intravenous administration, although transgene expression was efficiently observed in liver and spleen, the expression was rapidly decreased. In addition, second administration generated far weaker transgene expression likely due to immune response and facilitated generation of neutralizing antibody. Although neutralizing antibody was not raised in intragastric administration, the efficiency of transgene expression was weak. In intranasal administration, transgene was efficiently expressed and neutralizing antibody was not significantly raised. Consequently, second administration gave similar transgene expression to first one. In intramuscular administration, although the transgene was efficiently expressed, secondary administration gave significantly lower expression, which also decreased much faster, likely due to immune response. However, use of HAdV-5 at third administration, it induced equivalent expression to first administration of SAdV1-GFluc likely due to difference of immunogenicity. Similar results were obtained in opposite order of inoculation. From these results, the authors conclude that intranasal and intramuscular administration are preferred routes to use SAdV-1, and that pre-existing immunity to HAdV-5 or immunity induced by repeated use of HAdV-5 are good reasons to use SAdV-1 vector.
Although human adenovirus has been used for virus vector for gene transfer and vulcanization, pre-existing immunity and induced immunity have been the problems. To solve this problems, simian adenovirus vector has become focus of interest because of absence of pre-existing immunity due to serotype difference. The authors’ results would provide valuable information for readers in the field to utilize the simian adenovirus vector. The manuscript could be improved by addressing points listed below.
Minor points.
1. In general, firefly luciferase is thought to be a relatively unstable protein. In Figure 2 F~I, in the lung, although virus copy number decreased, the luciferase activity stayed long. This reviewer cannot understand the discrepancy in the results. In theory, luciferase activity is expected to decrease in parallel with virus copy number, since luciferase is rather short-lived protein. In addition, in Figure 3 F~I, in the lung, the luciferase activity decreased at least for 14d. Instead, the luciferase activity stayed long in the brain. Was the luciferase activity properly measured within the linear range?
2. It would be kind for readers to provide information about homology of receptors such as CAR between the species. Are they homologous enough for applying the results obtained in mouse to human?
Round 2
Reviewer 1 Report
Comments and Suggestions for Authors
The authors sufficiently addressed my concerns.
Minor point:
Remove the numbering for animals in Figs. 2C, 3C, and 4C; it suggests that the same animals were imaged throughout the experiment.
Comments on the Quality of English Language
Minor editing is necessary.